# DNA methylation and single-nucleotide polymorphisms in DDX58 are associated with hand, foot and mouth disease caused by enterovirus 71

Ya-Ping Li[1], Chen-Rui Liu[1], Hui-Ling Deng[2,3], Mu-Qi Wang[1], Yan Tian[1], Yuan Chen[2], Yu-Feng Zhang[2], Shuang-Suo Dang[1]*, Song Zhai[1]*

1 Department of Infectious Diseases, Xi'an Jiaotong University Second Affiliated Hospital, Xi'an, China, 2 Department of Infectious Diseases, Xi'an Children's Hospital, Xi'an, China, 3 Department of Pediatric, Xi'an Central Hospital, Xi'an, China

* dangshuangsuo123@xjtu.edu.cn (S-SD); zhaisong1103@xjtu.edu.cn (SZ)

**Data Availability Statement:** All relevant data are within the manuscript and its Supporting Information files.

## Abstract

### Background

This research aimed to explore the association between the RIG-I-like receptor (RIG-I and MDA5 encoded by DDX58 and IFIH1, respectively) pathways and the risk or severity of hand, foot, and mouth disease caused by enterovirus 71 (EV71-HFMD). In this context, we explored the influence of gene methylation and polymorphism on EV71-HFMD.

### Methodology/Principal findings

60 healthy controls and 120 EV71-HFMD patients, including 60 mild EV71-HFMD and 60 severe EV71-HFMD patients, were enrolled. First, MiSeq was performed to explore the methylation of CpG islands in the DDX58 and IFIH1 promoter regions. Then, DDX58 and IFIH1 expression were detected in PBMCs using RT-qPCR. Finally, imLDR was used to detect DDX58 and IFIH1 single-nucleotide polymorphism (SNP) genotypes. Severe EV71-HFMD patients exhibited higher DDX58 promoter methylation levels than healthy controls and mild EV71-HFMD patients. DDX58 promoter methylation was significantly associated with severe HFMD, sex, vomiting, high fever, neutrophil abundance, and lymphocyte abundance. DDX58 expression levels were significantly lower in mild patients than in healthy controls and lower in severe patients than in mild patients. Binary logistic regression analysis revealed statistically significant differences in the genotype frequencies of DDX58 rs3739674 between the mild and severe groups. GeneMANIA revealed that 19 proteins displayed correlations with DDX58, including DHX58, HERC5, MAVS, RAI14, WRNIP1 and ISG15, and 19 proteins displayed correlations with IFIH1, including TKFC, IDE, MAVS, DHX58, NLRC5, TSPAN6, USP3 and DDX58.

**Funding:** This work was supported by the National Natural Science Foundation of China (grant no. 81701632, received by YPL). The funders had no role in study design, data collection and analysis, decision to publish, or preparation of the manuscript.

**Competing interests:** The authors declare that they have no competing interests.

## Conclusions/Significance

DDX58 expression and promoter methylation were associated with EV71 infection progression, especially in severe EV71-HFMD patients. The effect of DDX58 in EV71-HFMD is worth further attention.

### Author summary

EV71-HFMD is now prevalent in many Asia countries, including China, which caused a lot of death over the last few years. Therefore, it's important to distinguish patients with a severe tendency. In this research, we investigated the association between the epigenetic mechanisms of RIG-I-like pattern recognition receptors (DDX58 and IFIH1) and EV71-HFMD. It showed that children with higher level of DDX58 promoter methylation were related to severe EV71-HFMD and other clinical indicators. The DDX58 mRNA expression level was significantly lower in severe patients. Besides, DDX58 polymorphisms played an important role in the severity of EV71-HFMD. Therefore, various epigenetic biomarkers of DDX58 may be used as indicators to help clinicians identify EV71-HFMD patients with a tendency towards severity.

## Introduction

In recent years, hand, foot and mouth disease (HFMD), a highly infectious digestive tract disease that is mainly caused by enterovirus 71 (EV71) and coxsackievirus A16 (CA16) infection among children and infants, has been prevalent in China [1–3]. Although most patients with HFMD can recover, it was reported that during 2008~2015, there were more than 120,000 cases of severe HFMD in mainland China, of which more than 3300 died [4,5]. In the Asia-Pacific region, HFMD has also been prevalent in many other countries, such as Japan, South Korea, and Singapore [6–8]. Epidemiological investigations have shown that EV71 is still the main pathogen of severe HFMD, although other enteroviruses causing HFMD have increased in prevalence in recent years [9,10].

Retinoic acid-inducible gene I (RIG-I, encoded by the DDX58 gene) and melanoma differentiation-associated protein 5 (MDA5, encoded by the IFIH1 gene) are both RIG-I-like pattern recognition receptors (PRRs) that can be activated by double-stranded RNA (dsRNA) and interact with mitochondrial antiviral signaling (MAVS) proteins in cells, resulting in the activation and expression of type I interferon genes [9,10,11]. In other words, they are important receptors of the early immune response after virus infections. Recent studies have shown that hantavirus, rotavirus and filovirus as well as others can be specifically recognized by RIG-I and MDA5 [12–14], which cause antiviral effects. EV71 is an RNA virus, and dsRNA is produced during the process of its replication. It can be speculated that RIG-I and MDA5 may also be activated by EV71 to produce antiviral effects. A recent study showed that EV71 can activate MDA5 in human neural cells and be inhibited by inducing the expression of interferon (IFN)-β though the MDA5 signaling pathway [15]. However, some other studies showed that EV71 may inhibit the production of type I IFN by affecting the signaling pathway mediated by RIG-I and MDA5 [16,17]. Regardless of the circumstances, RIG-I and MDA5 are closely related to the prognosis of HFMD caused by EV71 (EV71-HFMD).

Base methylation and demethylation in the DNA promoter region are often considered important ways to regulate gene expression levels [18]. The degree of methylation is different

in different regions of DNA. The CpG context is highly methylated, while most CpG islands are not methylated, which means that CpG islands in the promoter region can be methylated, resulting in the inhibition of gene expression [19–22]. The gene expression of RIG-I and MDA5 is also regulated by the methylation level of CpG islands in the promoter region of the coding genes [23,24]. The methylation level of CpG islands can reflect the expression of RIG-I and MDA5 and indirectly reflect the antiviral effect [23]. Zhang et al found that the methylation level of the chicken genome was significantly upregulated after infection with H5N1 avian influenza virus [25]. Zheng Zhu et al. showed that EV71 infection may change the methylation status of CpG islands in host DNA and regulate gene expression [26]. However, the antiviral effect is not only associated with regulated gene expression levels but also related to many other parameters, among which single-nucleotide polymorphisms (SNPs) and gene mutations are important research focuses. Our previous study indicated that RIG-1 (rs3739674 and rs9695310) polymorphisms were associated with an increased risk of EV71-HFMD in Chinese children, whereas the RIG-1 rs3739674 and TLR3 rs5743305 polymorphisms were associated with disease severity [27]. In this study, we investigated the association among the RIG-I and MDA5 promoter region methylation, nucleic acid sequence mutation and EV71-HFMD severity.

## Methods

### Ethics statements

This research was approved by the Medical Ethics Committee of the study hospital. Written informed consent was obtained from the parent/guardian of each participant under 18 years of age.

### Study population

Clinical specimens used in this research were collected from hospitalized HFMD patients during the 2017–2020 peak season (April-July and October-November) of HFMD in Xi'an at Xi'an Jiaotong University Second Affiliated Hospital and Xi'an Children's Hospital. HFMD was diagnosed based on criteria in the Hand, Foot and Mouth Disease Clinical Guidelines (2018 edition), which was issued by the National Health and Family Planning Commission of the People's Republic of China. Children with mild HFMD had rashes on their hands, feet, mouths, and buttocks, with or without fever. Severe case had one of the following complications involving nervous system, respiratory system and circulatory system: aseptic meningitis, encephalitis, acute flaccid paralysis, pulmonary edema or hemorrhage, and cardiopulmonary collapse. EV71 RNA detection in peripheral blood of all participants were positive. HFMD cases, which were positive for other enteroviruses or co-detected with other viral infections, were excluded.

Another group of 60 normal children served as the control group to collect blood samples with the informed consent of the children's guardians, which were enrolled from the healthy infants of same age for physical examination at the same hospital and the same time as well as the patients group. The details of the control group were displayed in S1 Table.

### Detection of EV71 infection in the clinical laboratory

EV71 detection in stool/throat specimens in the study hospital was primarily performed using commercially available panenterovirus, EV71, and CA16 diagnostic kits (Da An Gene Co., Ltd., Guangzhou, China) according to the manufacturer's instructions. If the quick diagnosis was EV71 positive, viral RNA was extracted from serum using an RNA extraction kit (Qiagen,

Germany). EV71 infection in patients was confirmed by positive EV71 identification through reverse transcription-PCR (RT-PCR) detection of the virus nucleic acid in serum.

## Collection and preservation of specimens

For children with EV71-HFMD, a total of 120 Chinese patients, including 60 mild and 60 severe patients, were included in this study. Venous blood samples of all participants were collected in the morning after overnight fasting, placed in a sterile, enzyme-free EDTA (1 ml) anticoagulation tube and a PAXgene Blood RNA Tube (2 ml), and stored at −30°C and −80°C for extraction of genomic DNA and RNA, respectively.

## DNA extraction and methylation analysis

Genomic DNA was extracted from enzyme-free EDTA-blood samples using a Gentra Puregene Blood Kit (Germany) according to the manufacturer's instructions. CpG islands located in the DDX58 and IFIH1 promoter regions were selected for analysis. Bisulfite conversion of 1 μg of genomic DNA was performed with an EZ DNA Methylation-GOLD Kit (Zymo Research, CA, USA). After PCR amplification (HotStarTaq Polymerase Kit, TAKARA, Tokyo, Japan) of target CpG regions and library construction, the products were sequenced on an Illumina MiSeq Benchtop Sequencer (CA, USA) with a mean coverage of >600X. The methylation level at each CpG site, named based on the relative distance to the transcriptional start site (TSS) in base pairs (bp), was calculated as the percentage of methylated cytosines divided by the total tested cytosines.

## Quantitative real-time PCR (qrt-PCR)

Total RNA was extracted from the white blood cells of 120 EV71-HFMD patients and 60 normal children. Using SYBR Advantage qPCR Premix (Takara), qRT-PCR to investigate the expression of DDX58 and IFIH1 was carried out on an ABI 7300 Real-Time PCR System (Applied Biosystems). Glyceraldehyde-3-phosphate dehydrogenase (GAPDH) served as the housekeeping gene for standardization. GAPDH expression were same in the groups. All assays were performed at least in triplicate, and the relative gene expression is presented as $2^{-\Delta\Delta Ct}$. The primer pairs utilized in this study were as follows: DDX58-Forward (AGACCC TGGACCCTACCTACA) and DDX58-Reverse (CTCCATTGGGCCCTTGTTGT) and IFIH1-Forward (ATTCAGGCACCATGGGAAGTG) and IFIH1-Reverse (TTTGGTAAGGCCTGAGCTGGA).

## SNP genotyping

Genomic DNA was extracted using a Gentra Puregene Blood Kit (Germany) from enzyme-free EDTA-blood samples according to standard protocols. Based on our previous research, positive SNP sites associated with EV71 infection were selected. DDX58 SNP (rs3739674/5'-UTR exon 1) and IFIH1 SNP (rs1990760/nonsynonymous exon 15) genotyping was performed using an improved multitemperature ligase detection reaction (imLDR) technique (Genesky Biotechnologies, Shanghai, China). Specifically, imLDR determined genotypes by PCR amplification of genomic DNA and sequencing. PCR primers were separately designed using Primer 6.0 software and synthesized by Shanghai Sangon Biotech Company. The primer sequences were as follows: rs3739674F: CCTAAACCAGGGGGCCATGTAG, rs3739674R: TCCTCGGAAAATCCCTGCTTTC; rs1990760F: GGCCCACAGCAATTTACTCACC, and rs1990760R: GCAATAACAAGCCTGGGAAGCA.

### DDX58 and IFIH1 network analysis

To derive *DDX58* and *IFIH1* functions, GeneMANIA (http://www.genemania.org/) was used to obtain genes with functions similar to those of *DDX58* and *IFIH1* and generate interactive functional association networks (Warde-Farley, 2010). Finally, the *DDX58* and *IFIH1* networks was built, including colocalization, coexpression, physical interactions, genetic interaction, and predictions, as well as their functions.

### Statistics

Statistical analyses were performed with SPSS 22.0 software (SPSS Inc., Chicago, IL). The methylation and expression levels among the three groups were not normally distributed, so these variables were summarized as medians (interquartile ranges, IQRs). Then, a nonparametric test was used for comparisons of continuous variables, and significance of differences between groups was analyzed by the Mann-Whitney test or Kruskal-Wallis test. The heatmap ("heatmap.2" R package) was constructed on the average methylation expression after standardization in order to construct separate methylation correlation matrices for disease status. The correlation between the average methylation level and expression of each gene was assessed using Spearman rank correlation coefficient. The receiver operating characteristic (ROC) curves, areas under the ROC curve (AUCs), or partial AUCs as well as their 95% confidence intervals were calculated using the R package pROC. A Kruskal-Wallis test was applied to investigate significant changes in methylation levels depending on disease condition. False Discovery Rate (FDR) analysis were adjusted P-value. Genotype frequencies were analyzed by the Hardy-Weinberg test. The various genotype and allele frequencies were compared using binary logistic regression analysis between the EV71-HFMD and normal groups and between the mild case and severe case groups to calculate the $\chi 2$, P, and odds ratio (OR) values and 95% CI (adjusted by age and sex). For statistical analysis, GraphPad Prism version 7 was used. The significance level was set at $P < 0.05$.

## Results

### Clinicopathological characteristics

A total of 120 EV71-HFMD patients were recruited, with a median age of 35(15–61) months (range: 6–96 months). 80 patients were male, and 40 were female. Age- and sex-matched healthy controls (60 children) were also recruited in the study, with a median age of 35(27–66) months (range: 24–79 months) (S1 Table).

### DDX58 and IFIH1 methylation levels

To determine whether the DDX58 and IFIH1 promoters experience aberrant methylation patterns in EV71-HFMD patients, three groups, including 60 mild and 60 severe cases and 60 healthy controls, were selected for MethylTarget assays. Twenty-one CpG islands adjacent to the DDX58 promoter regions were sequenced (Table 1), and twenty-three CpG islands adjacent to the IFIH1 promoter regions were sequenced (Table 2). The results demonstrate that severe EV71-HFMD patients exhibit higher levels of DDX58 promoter methylation than healthy controls and mild EV71-HFMD patients (Fig 1A). To better characterize the methylation of DDX58, we also assessed the methylation level of each CpG site. We observed that the 29, 134, 154, 168, 232, 246 and 248 CpG sites of DDX58 had a significantly higher methylation pattern in severe HFMD patients than in the corresponding healthy controls and mild HFMD patients, and 179 CpG site had a significantly higher methylation pattern in healthy controls

**Table 1. Details of the CpG regions in the CpG islands of DDX58.**

| Position | Chr | Genome position | Distance 2TSS | Healthy | Mild | Severe | *P* | *FDR* |
|---|---|---|---|---|---|---|---|---|
| 29 | 9 | 32526145 | 177 | 0.006(0.005–0.008) | 0.007(0.006–0.010) | 0.009(0.008–0.010) | <0.001*** | <0.001*** |
| 32 | 9 | 32526148 | 174 | 0.007(0.006–0.010) | 0.008(0.007–0.009) | 0.007(0.006–0.009) | 0.327 | 0.450 |
| 39 | 9 | 32526155 | 167 | 0.006(0.005–0.008) | 0.006(0.005–0.009) | 0.007(0.005–0.009) | 0.220 | 0.304 |
| 51 | 9 | 32526167 | 155 | 0.008(0.006–0.009) | 0.007(0.006–0.009) | 0.008(0.006–0.010) | 0.299 | 0.619 |
| 74 | 9 | 32526190 | 132 | 0.005(0.004–0.006) | 0.005(0.004–0.007) | 0.005(0.003–0.006) | 0.070 | 0.177 |
| 81 | 9 | 32526197 | 125 | 0.009(0.007–0.011) | 0.009(0.007–0.010) | 0.008(0.007–0.010) | 0.629 | 0.688 |
| 110 | 9 | 32526226 | 96 | 0.008(0.007–0.010) | 0.007(0.005–0.010) | 0.008(0.006–0.010) | 0.144 | 0.216 |
| 125 | 9 | 32526241 | 81 | 0.006(0.005–0.008) | 0.006(0.005–0.008) | 0.007(0.005–0.008) | 0.633 | 0.688 |
| 130 | 9 | 32526246 | 76 | 0.006(0.005–0.008) | 0.007(0.005–0.009) | 0.008(0.005–0.009) | 0.095 | 0.153 |
| 134 | 9 | 32526250 | 72 | 0.005(0.004–0.007) | 0.006(0.004–0.008) | 0.007(0.006–0.009) | <0.001*** | <0.001*** |
| 154 | 9 | 32526270 | 52 | 0.009(0.007–0.010) | 0.009(0.007–0.010) | 0.012(0.010–0.017) | <0.001*** | <0.001*** |
| 168 | 9 | 32526284 | 38 | 0.010(0.009–0.011) | 0.008(0.007–0.010) | 0.010(0.009–0.012) | 0.003** | 0.055 |
| 172 | 9 | 32526288 | 34 | 0.008(0.007–0.010) | 0.008(0.006–0.010) | 0.007(0.005–0.009) | 0.051 | 0.161 |
| 174 | 9 | 32526290 | 32 | 0.008(0.006–0.010) | 0.007(0.006–0.010) | 0.008(0.007–0.010) | 0.456 | 0.688 |
| 179 | 9 | 32526295 | 27 | 0.008(0.006–0.010) | 0.006(0.005–0.008) | 0.007(0.006–0.009) | 0.033* | 0.143 |
| 217 | 9 | 32526333 | -11 | 0.007(0.005–0.008) | 0.008(0.006–0.009) | 0.007(0.006–0.008) | 0.087 | 0.153 |
| 232 | 9 | 32526348 | -26 | 0.006(0.005–0.007) | 0.007(0.006–0.009) | 0.010(0.009–0.011) | <0.001*** | <0.001*** |
| 235 | 9 | 32526351 | -29 | 0.006(0.005–0.007) | 0.006(0.004–0.007) | 0.006(0.005–0.007) | 0.864 | 0.888 |
| 239 | 9 | 32526355 | -33 | 0.006(0.005–0.007) | 0.006(0.005–0.008) | 0.006(0.005–0.008) | 0.181 | 0.277 |
| 246 | 9 | 32526362 | -40 | 0.009(0.007–0.010) | 0.009(0.007–0.011) | 0.012(0.009–0.014) | <0.001*** | <0.001*** |
| 248 | 9 | 32526364 | -42 | 0.007(0.005–0.009) | 0.008(0.006–0.009) | 0.011(0.009–0.014) | <0.001*** | <0.001*** |
| Average | | | | | | | | |
| Mean±SD | | | | 0.007±0.001 | 0.008±0.001 | 0.009±0.001 | <0.001*** | / |
| Median (IQR) | | | | 0.007(0.007~0.008) | 0.008(0.007~0.008) | 0.008(0.008~0.009) | <0.001*** | <0.001*** |

Chr, chromosome; Genome position, the position of the product on the reference genome; TSS, the mRNA transcription initiation site; Distance2TSS, the distance from the product to the TSS; Target strand, the product orientation. Specific CpG site methylation levels of the healthy, mild and severe groups are shown as the median (IQR, interquartile range). *P*: Kruskal-Wallis test. False Discovery Rate (FDR) analysis were adjusted P-value.

* *P*<0.05

** *P*<0.01

*** *P*<0.001.

than in the severe and mild HFMD patients(Table 1, Fig 2A). After adjust for p value by FDR, 29, 134, 154, 232, 246 and 248 CpG sites of DDX58 are still positive (Table 1, Fig 2A).

The average methylation levels of IFIH1 among the three groups were not significantly different (Fig 1C and 1D), but we also assessed the methylation level of each CpG site. We observed that the 50, 86 CpG sites, and the average methylation level of 50 CpG sites was significantly higher, while 86 CpG sites was significantly lower in severe HFMD patients than that in the corresponding healthy controls and mild HFMD patients (Table 2, Fig 2B).

## The association between DDX58 and IFIH1 methylation patterns and clinical indicators of EV71-HFMD

A ROC curve was used to assess the clinical application of DNA methylation for the prediction of EV71-HFMD severity. DDX58 methylation status was able to differentiate between mild and severe EV71-HFMD (AUC = 0.806 on average, AUC = 0.699 at the 29 CpG site, 0.660 at the 134 CpG site, 0.798 at the 154 CpG site, 0.903 at the 232 CpG site, 0.779 at the 246 CpG site

**Table 2. Details of the CpG regions in the CpG islands of IFIH1.**

| Position | Chr | Genome position | Distance 2TSS | Healthy | Mild | Severe | P | FDR |
|---|---|---|---|---|---|---|---|---|
| 27 | 2 | 163174873 | 345 | 0.007(0.006–0.009) | 0.007(0.005–0.010) | 0.006(0.005–0.008) | 0.051 | 0.318 |
| 45 | 2 | 163174891 | 327 | 0.006(0.005–0.008) | 0.006(0.005–0.008) | 0.006(0.005–0.009) | 0.913 | 0.567 |
| 50 | 2 | 163174896 | 322 | 0.010(0.008–0.012) | 0.0100.007–0.012) | 0.011(0.009–0.012) | 0.010* | 0.037* |
| 57 | 2 | 163174903 | 315 | 0.010(0.008–0.013) | 0.010(0.009–0.013) | 0.010(0.009–0.011) | 0.341 | 0.318 |
| 86 | 2 | 163174932 | 286 | 0.011(0.009–0.012) | 0.011(0.009–0.015) | 0.009(0.008–0.011) | 0.002** | 0.037* |
| 106 | 2 | 163174952 | 266 | 0.005(0.004–0.006) | 0.006(0.004–0.007) | 0.005(0.004–0.007) | 0.795 | 0.865 |
| 132 | 2 | 163174978 | 240 | 0.009(0.007–0.011) | 0.009(0.006–0.010) | 0.010(0.006–0.013) | 0.700 | 0.566 |
| 135 | 2 | 163174981 | 237 | 0.008(0.006–0.009) | 0.006(0.005–0.008) | 0.007(0.006–0.010) | 0.072 | 0.174 |
| 137 | 2 | 163174983 | 235 | 0.010(0.009–0.013) | 0.010(0.008–0.012) | 0.009(0.008–0.011) | 0.146 | 0.318 |
| 139 | 2 | 163174985 | 233 | 0.007(0.005–0.010) | 0.007(0.006–0.010) | 0.008(0.006–0.009) | 0.846 | 0.865 |
| 151 | 2 | 163174997 | 221 | 0.010(0.008–0.012) | 0.009(0.008–0.012) | 0.009(0.008–0.011) | 0.340 | 0.531 |
| 173 | 2 | 163175019 | 199 | 0.006(0.004–0.008) | 0.007(0.005–0.009) | 0.007(0.005–0.009) | 0.118 | 0.318 |
| 188 | 2 | 163175034 | 184 | 0.014(0.012–0.016) | 0.013(0.011–0.018) | 0.012(0.010–0.016) | 0.698 | 0.567 |
| 193 | 2 | 163175039 | 179 | 0.010(0.007–0.012) | 0.009(0.007–0.011) | 0.009(0.007–0.010) | 0.102 | 0.309 |
| 196 | 2 | 163175042 | 176 | 0.007(0.005–0.009) | 0.007(0.005–0.009) | 0.007(0.005–0.007) | 0.221 | 0.316 |
| 198 | 2 | 163175044 | 174 | 0.044(0.036–0.047) | 0.044(0.036–0.051) | 0.046(0.042–0.052) | 0.066 | 0.118 |
| 200 | 2 | 163175046 | 172 | 0.012(0.011–0.015) | 0.013(0.010–0.014) | 0.013(0.011–0.015) | 0.559 | 0.863 |
| 206 | 2 | 163175052 | 166 | 0.013(0.010–0.016) | 0.013(0.011–0.016) | 0.013(0.012–0.015) | 0.857 | 0.905 |
| 208 | 2 | 163175054 | 164 | 0.007(0.006–0.010) | 0.008(0.007–0.010) | 0.008(0.006–0.010) | 0.357 | 0.863 |
| 221 | 2 | 163175067 | 151 | 0.007(0.005–0.009) | 0.006(0.005–0.007) | 0.006(0.005–0.007) | 0.053 | 0.118 |
| 225 | 2 | 163175071 | 147 | 0.010(0.008–0.012) | 0.011(0.008–0.013) | 0.011(0.008–0.013) | 0.212 | 0.319 |
| 232 | 2 | 163175078 | 140 | 0.008(0.007–0.011) | 0.008(0.007–0.011) | 0.008(0.007–0.011) | 0.989 | 0.990 |
| 234 | 2 | 163175080 | 138 | 0.006(0.006–0.008) | 0.007(0.006–0.010) | 0.007(0.006–0.009) | 0.439 | 0.865 |
| Average | | | | | | | | |
| Mean±SD | | | | 0.010±0.001 | 0.010±0.001 | 0.011±0.001 | 0.536 | / |
| Median (IQR) | | | | 0.011(0.010–0.011) | 0.011(0.010–0.011) | 0.011(0.010–0.012) | 0.898 | 0.908 |

Chr, chromosome; Genome position, the position of the product on the reference genome; TSS, the mRNA transcription initiation site; Distance2TSS, the distance from the product to the TSS; Target strand, the product orientation. Specific CpG site methylation levels of the healthy, mild and severe groups are shown as the median (IQR, interquartile range). *P*: Kruskal-Wallis test. False Discovery Rate (FDR) analysis were adjusted P-value.

* *P*<0.05

** *P*<0.01

*** *P*<0.001.

and 0.860 at the 248 CpG site) (Fig 3A). The results suggest that DDX58 promoter methylation may be a useful marker to identify EV71-HFMD severity. Then, the relationship between DDX58 methylation and clinical features was analyzed. All EV71-HFMD samples were divided into two groups according to their median DDX58 methylation value (Table 3). Next, nine clinical indicators were analyzed by a χ2 test. We found that altered DDX58 promoter methylation was significantly associated with severe HFMD, sex, vomiting, high fever, neutrophil abundance, and lymphocyte abundance (Table 3). Specifically, a higher level of DDX58 promoter methylation was related to severe EV71-HFMD, increased neutrophil abundance and glucose levels, and decreased lymphocyte abundance.

The IFIH1 average methylation level was not significantly different among the three groups (Table 2), but there were three positive sites that were significantly different between mild and severe EV71-HFMD (AUC = 0.644 at the 50 CpG site, and 0.670 at the 86 CpG site) (Fig 3B). The clinical indicator statistical findings showed that average IFIH1 promoter methylation

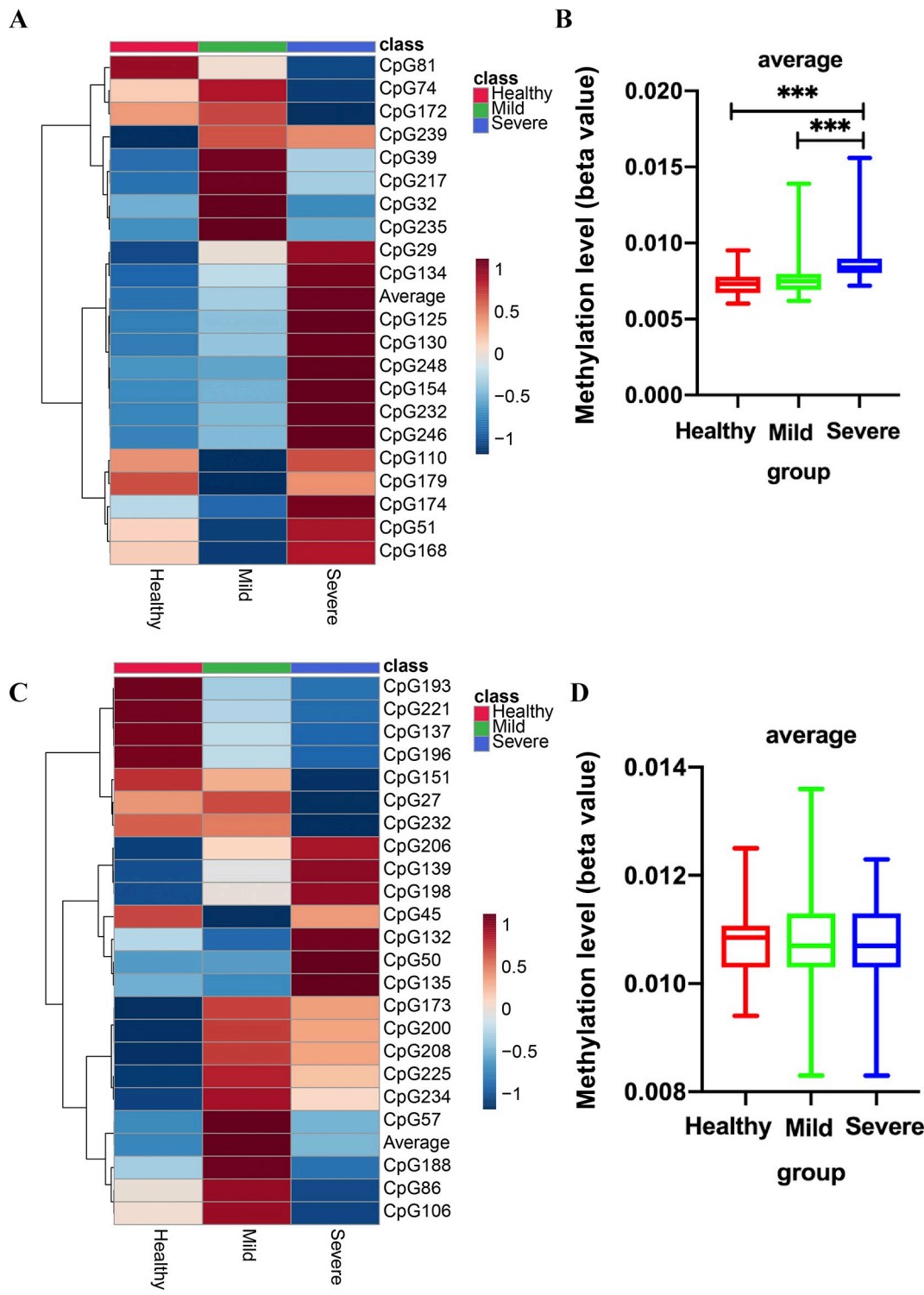

**Fig 1. The methylation patterns of DDX58 and IFIH1 in EV71-HFMD patients and healthy controls.** Each colored cell on the heatmap corresponds to an average concentration value in different groups. The methylation patterns of DDX58 in EV71-HFMD patients and healthy controls (A). DDX58 methylation levels of all samples, including healthy subjects and EV71-HFMD patients (B). The methylation patterns of IFIH1 in EV71-HFMD patients and healthy controls. (C). IFIH1 methylation levels of all samples, including those of healthy subjects and EV71-HFMD patients (D). ***$p < 0.001$.

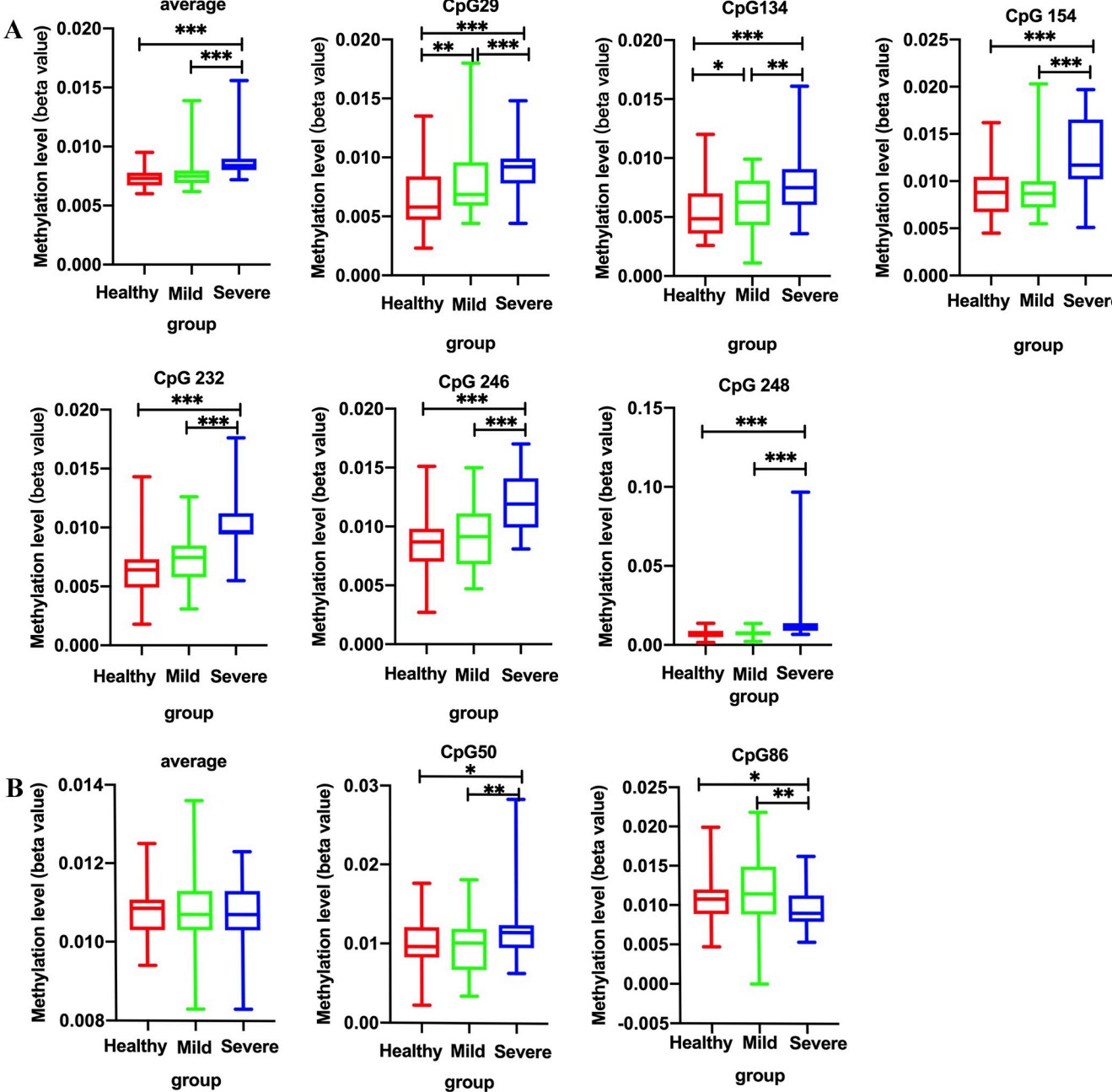

**Fig 2.** DDX58 methylation levels of each positive CpG site (the 29, 134, 154, 232, 246 and 248 CpG sites) in all samples, including healthy subjects and EV71-HFMD patients (A). IFIH1 methylation levels of each positive CpG site (the 50 and 86 CpG sites) in all samples, including those of healthy subjects and EV71-HFMD patients. (B) $^*p < 0.05$, $^{**}p < 0.01$, $^{***}p < 0.001$.

was not associated with disease severity, neutrophil abundance, glucose levels or lymphocyte abundance (Table 3).

## Determination of DDX58 and IFIH1 mrna expression

GAPDH is used as an internal control. DDX58 mRNA expression levels were significantly lower in mild patients (median 0.123, IQR 0.101–0.134) than in healthy controls (median

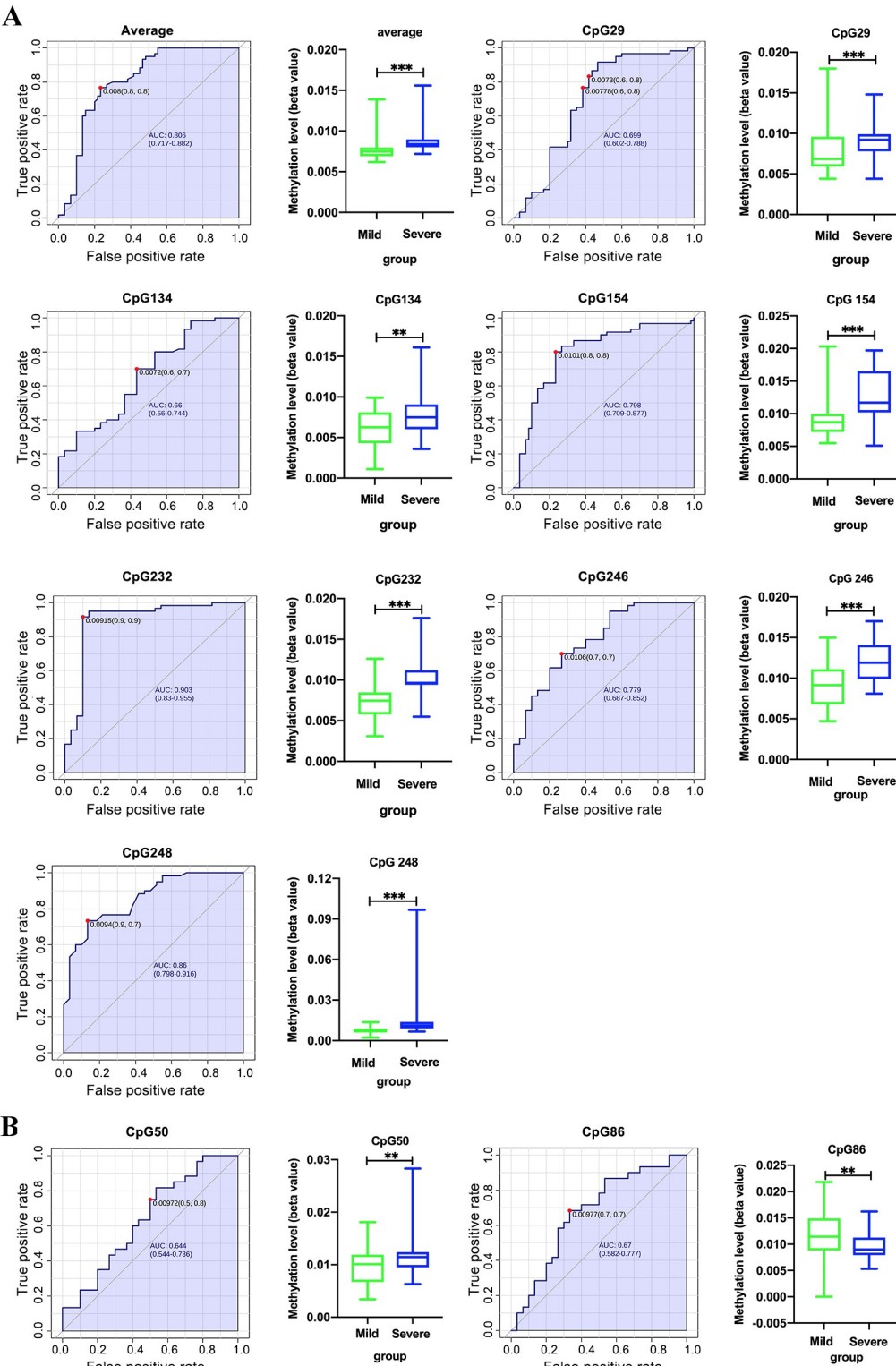

**Fig 3. DDX58 methylation status and levels of each positive CpG site (the 29, 134, 154, 232, 246 and 248 CpG sites) between mild and severe EV71-HFMD patients.** (A). IFIH1 methylation status and levels of each positive CpG site (the 50 and 86 CpG sites) between mild and severe EV71-HFMD patients. (B) $^*p < 0.05$, $^{**}p < 0.01$, $^{***}p < 0.001$.

**Table 3.** The relationship between the DDX58/IFIH1 methylation levels and clinicopathological factors in EV71-HFMD.

| Variable | DDX58 Average methylation (median as cutoff) | | | IFIH1 Average methylation (median as cutoff) | | |
|---|---|---|---|---|---|---|
| | Low | High | *P* | Low | High | *P* |
| Mild/severe | 45/15 | 15/45 | <0.001 | 32/33 | 28/27 | 0.855 |
| Male/female | 33/27 | 47/13 | 0.007 | 37/28 | 43/12 | 0.014 |
| Age (≤3/>3 years) | 34/26 | 45/15 | 0.034 | 48/17 | 31/24 | 0.044 |
| Vomiting (No/Yes) | 48/12 | 24/36 | <0.001 | 39/26 | 33/22 | 0.990 |
| Fever course (≤3/>3 days) | 19/41 | 14/46 | 0.307 | 18/47 | 15/40 | 0.959 |
| High fever (≤39/>39°C) | 30/30 | 11/49 | <0.001 | 21/44 | 20/35 | 0.641 |
| Neutrophil% | 45.74 (29.20–61.00) | 70.90 (48.70–75.65) | <0.001 | 57.10 (34.95–71.00) | 57.10 (41.30–75.40) | 0.176 |
| Glucose (mmol/L) | 6.00 (4.06–7.10) | 5.50 (4.41–6.80) | 0.990 | 5.50 (3.98–6.68) | 6.00(4.90–7.30) | 0.049 |
| Lymphocyte% | 53.40 (41.40–61.60) | 32.00 (19.60–37.60) | <0.001 | 35.40 (24.60–55.65) | 44.50 (20.30–55.80) | 0.918 |
| mRNA expression /GAPDH | 0.120 (0.099–0.136) | 0.100 (0.088–0.116) | <0.001 | 0.039 (0.003–0.056) | 0.037(0.026–0.054) | 0.246 |

Neutrophil%, Glucose, Lymphocyte% and mRNA expression were presented as Median (interquartile range, IQR); P: Mann-Whitney test.

0.184, IQR 0.149–0.206, $P<0.001$; Fig 4A) and lower in severe patients (median 0.098, IQR 0.088–0.112) than in mild patients (median 0.123, IQR 0.101–0.134, $P<0.001$; Fig 4C). Additionally, DDX58 mRNA levels were able to differentiate between mild and severe EV71-HFMD (AUC = 0.814 on average; Fig 4B). IFIH1 mRNA expression levels were significantly lower in severe patients (median 0.035, IQR 0.027–0.040) than in mild patients (median 0.043, IQR 0.036–0.060, $P = 0.001$; Fig 4F), and IFIH1 mRNA levels were able to differentiate between mild and severe EV71-HFMD (AUC = 0.669 on average; Fig 4E).

## Promoter methylation is associated with DDX58 expression

The relationship between the level of DNA methylation in the promoter region of DDX58 and the expression of DDX58 mRNA was analyzed. Spearman's correlation analysis showed that the average level of CpG methylation in the DDX58 promoter region was negatively correlated with the level of DDX58 mRNA expression among the three groups. Furthermore, the 29, 130, 134, 154, 232, 246, 248 CpG sites and the average level of CpG methylation were negatively correlated with DDX58 mRNA expression in all three groups (Fig 5A). Between the mild and severe groups, the 154, 179, 232, 248 CpG sites and the average level of CpG methylation in the DDX58 promoter region were negatively correlated with the level of DDX58 mRNA expression (Fig 5B). In addition, there was no correlation between IFIH1 expression and the average methylation level (Fig 5C and 5D).

Simple linear regression analysis were performed to show the variance in DDX58 expression accounted by HFMD severity (R square = 0.293, p<0.001), by DDX58 methylation (R square = 0.058, p = 0.008), and the variance in IFIH1 expression accounted by HFMD severity (R square = 0.033, p = 0.046), by IFIH1 methylation (R square = 0.007, p = 0.347).

## Relationship between DDX58 polymorphisms and severity of EV71-HFMD

After adjusting for sex and age, there were significant differences in genotype frequencies of DDX58 rs3739674 identified through binary logistic regression analysis between the mild and severe groups according to an allele model for the C allele (G vs. C: OR 1.421, 95% CI 1.077–1.873, $P = 0.013$) (Table 4). *DDX58* mRNA expression and methylation levels were determined for each genotype at polymorphic sites (Fig 6A and 6B). At *DDX58* SNP rs3739674, genotype GG (median 0.112, IQR 0.094–0.133) was associated with significantly lower *DDX58* mRNA

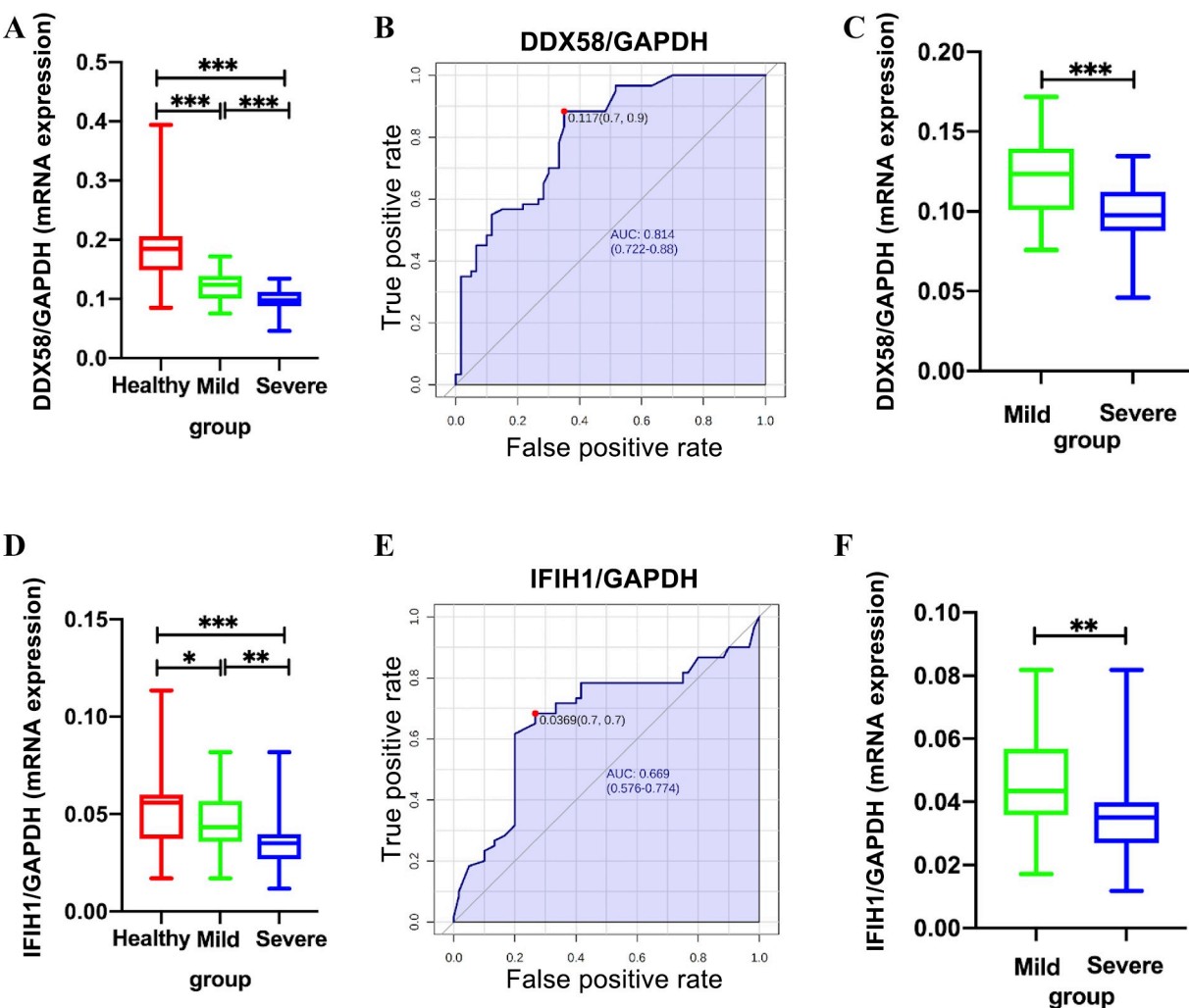

**Fig 4.** Determination of DDX58 mRNA expression in all samples (A). AUC of DDX58 mRNA levels between mild and severe EV71-HFMD patients (B). Determination of DDX58 mRNA expression between mild and severe EV71-HFMD patients (C). Determination of IFIH1 mRNA expression in all samples (D). AUC of IFIH1 mRNA levels between mild and severe EV71-HFMD patients (E). Determination of IFIH1 mRNA expression between mild and severe EV71-HFMD patients (F). $***p < 0.001$.

expression levels than both the CC (median 0.146, IQR 0.010–0.185) and GC genotypes (median 0.132, IQR 0.106–0.161, Fig 6A). At *DDX58* SNP rs3739674, genotype GG (median 0.009, IQR 0.008–0.009) was associated with significantly higher *DDX58* methylation levels than both the CC (median 0.007, IQR 0.006–0.007) and GC genotypes (median 0.008, IQR 0.007–0.008, Fig 6B).

After adjusting for sex and age, there were no significant differences in genotype frequencies of IFIH1 rs1990760 among the mild and severe groups (Table 4). *IFIH1* mRNA expression and methylation levels were determined for each genotype at polymorphic sites (Fig 6C and 6D). At *IFIH1* SNP rs1990760, genotype TT (median 0.012, IQR 0.011–0.013) was associated with significantly higher *IFIH1* methylation levels than both the CC (median 0.010, IQR 0.009–0.011) and CT genotypes (median 0.011, IQR 0.010–0.012, Fig 6D). In addition, there was no correlation between IFIH1 expression and the genotypes of *IFIH1* SNP rs1990760 (Fig 6C).

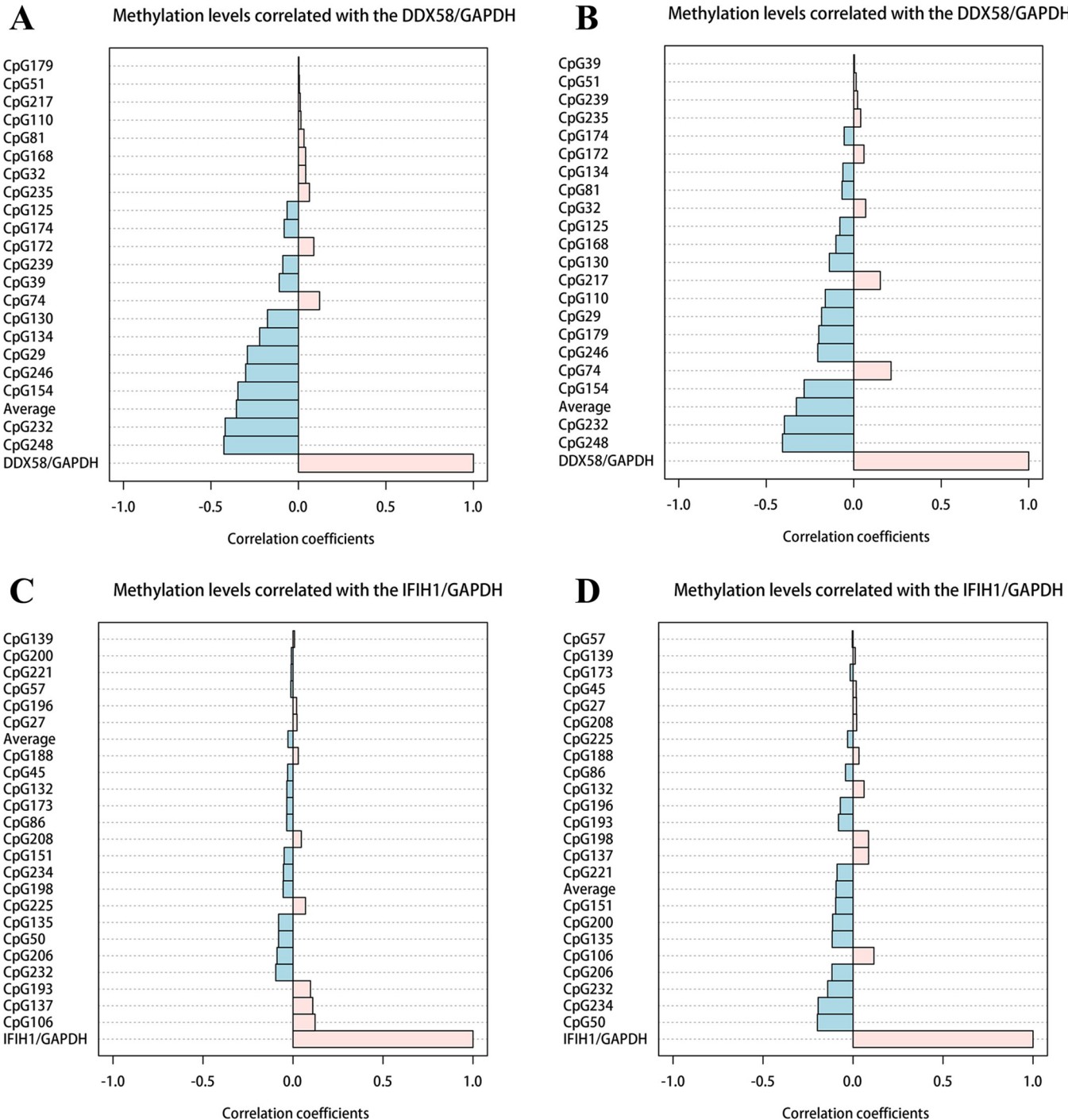

**Fig 5.** Spearman rank correlation coefficient analysis was performed between DDX58 and IFIH1 methylation and mRNA expression. All samples for DDX58 (A). EV71-HFMD patients for DDX58 (B). All samples for IFIH1 (C). EV71-HFMD patients for IFIH1 (D).

## DDX58 and IFIH1 network analysis

A gene–gene interaction network for DDX58 was constructed, and its function was analyzed using the GeneMANIA database (Fig 7A). GeneMANIA revealed that 19 proteins displayed correlations with DDX58, including DHX58, HERC5, MAVS, RAI14, WRNIP1, and ISG15.

**Table 4. DDX58 rs3739674 and IFIH1 rs1990760 genotype distribution and allele frequencies in the mild and severe HFMD groups.**

| Polymorphism | Genotype | Mild cases (%) (n = 60) | Severe cases (%) (n = 60) | P | OR(95% CI) |
|---|---|---|---|---|---|
| rs3739674 | Codominant model | | | | |
| | CC | 19 (31.7) | 6 (10.0) | 1.000 (ref.) | |
| | GC | 20 (33.3) | 22 (36.7) | **0.028** | **3.906 (1.163–13.122)** |
| | GG | 21(35.0) | 32(53.3) | **0.010** | **2.083 (1.193–3.635)** |
| | Dominant model | | | | |
| | CC | 19 (31.7) | 6 (10.0) | 1.000 (ref.) | |
| | GC+GG | 41 (68.3) | 54 (90.0) | **0.007** | **3.670 (1.313–10.255)** |
| | Allele model | | | | |
| | C | 58 (48.3) | 34 (28.3) | 1.000 (ref.) | |
| | G | 62 (51.7) | 86 (71.7) | **0.013** | **1.421 (1.077–1.873)** |
| *H-W* | | 0.010 | 0.452 | | |
| rs1990760 | Codominant model | | | | |
| | CC | 37 (61.7) | 37 (61.7) | 1.000 (ref.) | |
| | CT | 17 (28.3) | 22 (36.7) | 0.548 | 1.289 (0.563–2.952) |
| | TT | 6 (10.0) | 1 (1.6) | 0.107 | 0.398 (0.129–1.222) |
| | Dominant model | | | | |
| | CC | 37 (61.7) | 37 (61.7) | 1.000 (ref.) | |
| | CT+TT | 23 (38.3) | 23 (38.3) | 0.947 | 1.027 (0.472–2.232) |
| | Allele model | | | | |
| | C | 91 (75.8) | 96 (80.0) | 1.000 (ref.) | |
| | T | 29 (24.2) | 24 (20.0) | 0.501 | 0.897 (0.653–1.231) |
| *H-W* | | 0.078 | 0.259 | | |

Data are presented as n (%). OR, odds ratio; 95% CI, 95% confidence interval. Bold values emphasize the positive locus determined by statistical analysis.

DDX58 was correlated with RAI14 in terms of physical interactions. Further functional analysis showed that these 20 proteins displayed the greatest correlation with negative regulation of type I IFN production (FDR = 1.08E-24), regulation of type I IFN production (FDR = 4.83E-19), type I IFN production (FDR = 4.83E-19), negative regulation of cytokine production (FDR = 8.24E-19), negative regulation of multicellular organismal process (FDR = 5.43E-15), and response to virus (FDR = 1.33E-13). Additionally, these proteins were correlated with positive regulation of IFN-β production, regulation of multiorganism process, positive

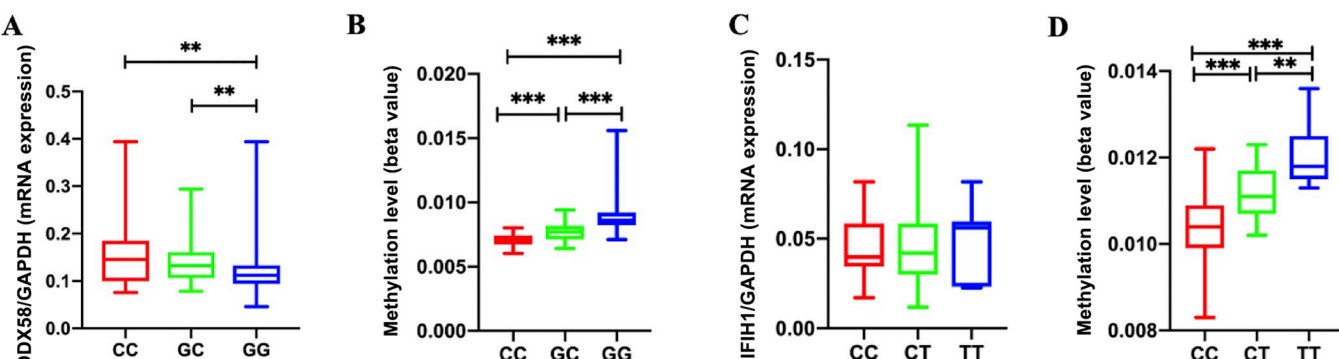

**Fig 6.** DDX58 *expression* levels in each genotype at polymorphic sites (A). *DDX58* methylation levels in each genotype at polymorphic sites (B). IFIH1 *expression* levels in each genotype at polymorphic sites (C). IFIH1 methylation levels in each genotype at polymorphic sites (D) **p < 0.01, ***p < 0.001.

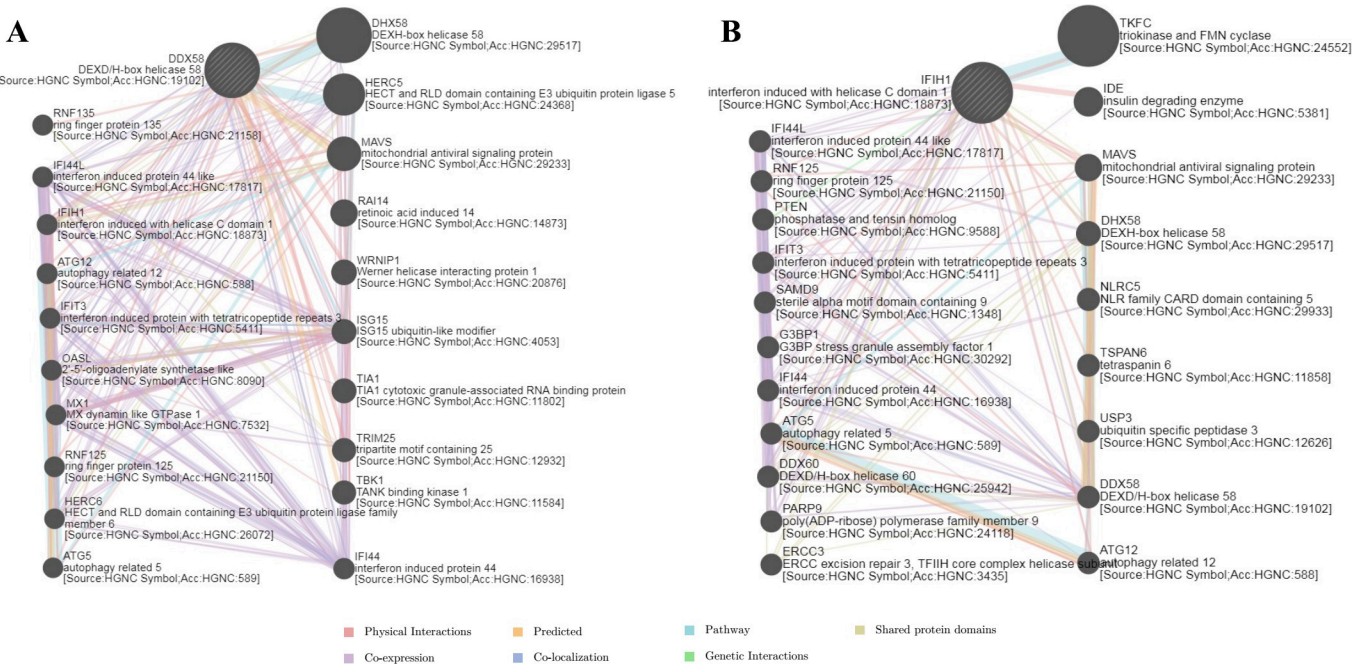

**Fig 7.** Biological interaction network of DDX58 (A) and IFIH1(B). Different colors represent diverse bioinformatics methods.

regulation of type I IFN production, regulation of IFN-β production, and defense response to virus. (S2 Table).

A gene–gene interaction network for IFIH1 was constructed, and its function was analyzed using the GeneMANIA database (Fig 7B). GeneMANIA revealed that 19 proteins displayed correlations with IFIH1, including TKFC, IDE, MAVS, DHX58, NLRC5, TSPAN6, USP3, and DDX58. IFIH1 was correlated with IDE in terms of physical interactions. Further functional analysis showed that these 20 proteins displayed the greatest correlation with negative regulation of type I IFN production (FDR = 2.22E-13), regulation of type I IFN production (FDR = 6.04E-10), type I IFN production (FDR = 6.04E-10), negative regulation of cytokine production (FDR = 7.74E-10), negative regulation of multicellular organismal process (FDR = 2.74E-07), response to virus (FDR = 7.44E-08), and regulation of innate immune response (FDR = 1.05E-06). Additionally, these proteins were correlated with the cytoplasmic PRR signaling pathway in response to virus, cellular response to virus, defense response to virus, defense response to other organisms, and single-stranded RNA binding. (S3 Table).

## Discussion

EV71 causes HFMD, neurological complications or even fatal diseases in young children and infants. It is still a major challenge due to some children experiencing serious consequences [28]. PRRs are the first line of defense against EV71 infection. DDX58 and IFIH1 are the main antiviral molecules among host innate immune PRRs. DDX58 and IFIH1 can distinguish different RNA viruses and play a key role in the host antiviral response [29]. Lack of both DDX58 and IFIH1 decreased innate immune signaling in the target cells of West Nile virus (WNV)-infected mice, and the pathogenesis was serious in vivo [30]. In addition, virus-infected cells can also be recognized by the molecular pattern recognition of DDX58 and IFIH1 [31]. DDX58 is essential in the signal transduction of influenza A virus, influenza B virus and human respiratory syncytial virus [32]. In contrast, IFIH1 uniquely triggers innate immune

defenses during picornavirus infections [29]. Our previous study confirmed the potential association between genetic polymorphisms and the risk and severity of EV71 infection, while the DDX58 rs9695310 GC/CC genotype increased the risk of EV71 infection [27]. Overexpression of the MDA5 protein reverses the suppression of IRF3 activation caused by EV71 infection, indicating that IFIH1 is an important factor for EV71 RNA-activated type I IFN expression [33]. The association between the MDA5 rs1990760 polymorphism and an increased risk of severe EV71 infection was indicated in Chinese children [34]. In view of these findings, we further analyzed the relationship between PRR gene methylation and SNPs and EV71 infection and tried to explore the reasons for different clinical outcomes.

An increasing number of studies have focused on the association between aberrant methylation and various diseases in the field of life science, such as cancer [35,36], diabetes [37] and cardiovascular diseases [38]. At the same time, many studies have shown that viral infection can affect the DNA methylation status of host cells, leading to changes in epigenetic patterns [39,40]. Through epigenetic mechanisms and host cell fusion, both Epstein-Barr virus and cytomegalovirus can demethylate the host syncytial protein 1 and 2 genes, increasing gene transcription and leading to the formation of syncytial bodies in tissues where those genes are normally hypermethylated and silenced [41,42]. In recent studies, epigenetic mechanisms seem to be an important part of the pathophysiology and disease severity of COVID-19 [43]. There are also many studies on the relationship between EV71 infection and methylation. Acute enterovirus infections, such as those caused by EV71 and CA16, induce significant changes in host cellular DNA methylation status. Differentially methylated CpG sites are widely distributed throughout the genome, which may affect the expression of many target genes and therefore contribute to disease occurrence [26]. Other studies have identified that EV71 infection increased methylation of the miR-17-92 promoter, while expression of the hsa-miR-17-92 cluster was significantly downregulated [44]. Our previous studies have shown that vitamin D receptor expression and promoter methylation are associated with the progression of EV71 infection [45]. The DNA methylation level of the VDR promoter in children with severe EV71- HFMD was lower than that in healthy controls and mild HFMD patients, and the average CG methylation level in the VDR promoter region was negatively correlated with the mRNA expression level of VDR in both EV71-HFMD patients and healthy controls.

In this study, low DDX58 expression is significantly associated with severe HFMD compared with mild HFMD or healthy controls. We found that hypermethylation levels may decrease the expression of DDX58 and increase susceptibility to EV71 infection. Further research also confirmed this hypothesis, which showed that the mRNA expression levels of DDX58 and IFIH1 in severe patients were significantly lower than those in mild patients and healthy controls. These results suggest that low expression of DDX58 may be associated with high methylation level and SNP typing. High DDX58 promotor methylation might be a reason for low DDX58 expression. Simple linear regression analysis shows that the variance in DDX58 expression could be 29.3% explained by the severity of HFMD, while IFIH1 expression 3.3%, suggesting that low expression of DDX58 can increase the susceptibility of children to EV71, accelerate the progress of HFMD, and lead to severe disease easily. Therefore, DDX58 expression plays a bigger role in HFMD severity. Correlation analysis and simple linear regression analysis between DDX58/IFIH1 expressions and their promotor CpG sites methylation suggest that promotor methylation might have a minimal effect on DDX58 expressions. These findings indicate that DDX58 and IFIH1 methylation are early events in EV71-HFMD. We selected EV71-HFMD patients as the research objects and only collected samples after infection. We selected healthy children as controls instead of the patients themselves before infection. In addition, there are many factors affecting the expression of DDX58. Therefore, we could not compare the data before and after the onset, nor can we find out the causal

relationship between DDX58 and severe HFMD, which is one of the limitations of our study. Maybe follow-up animal experiment could confirm the changes of DDX58 expression before and after infection and its impact on the severity of the disease.

In addition, some studies have shown that SNPs may affect the DNA methylation level of the CpG island region or regulate gene expression, thus participating in the occurrence and development of diseases [46]. SNP sites are significantly correlated with gene expression and methylation levels, and they can affect the methylation level of multiple CpG sites [47,48]. Our previous study showed that DDX58 rs3739674 and rs9695310 were associated with an increased risk of EV71-HFMD in Chinese children, while DDX58 rs3739674 and TLR3 rs5743305 were associated with the severity of the disease. These findings supported the important role of the innate immune mechanism in EV71 infection [27]. On this basis, we further collected another group of samples to confirm the correlation between SNPs, mRNA expression and methylation of DDX58 and IFIH1. We found that there was a significant difference in the DDX58 rs3739674 genotype frequency between the mild and severe EV71-HFMD groups. At DDX58 SNP rs3739674, the expression level of DDX58 mRNA associated with the GG genotype was significantly lower than that associated with the CC and GC genotypes, while the DDX58 methylation levels associated with the GG genotype were significantly higher than those associated with the other two genotypes. These results indicated that the DDX58 SNP rs3739674 GG genotype was associated with the severity of EV71 infection, which is consistent with our previous results. Many viruses interfere with and inhibit the RIG-I (DDX58) signaling pathway through their own proteins and nucleic acids, thereby inhibiting the cellular antiviral response. DDX58 rs3739674 is located in the 5′UTR, which may affect the transcription level and further regulate the protein expression level through mutation.

Similarly, the IFIH1 gene is located on chromosome 2q24.2 and encodes a DEAD box protein involved in immune regulation. Previous studies have shown that the IFIH1 rs1990760 polymorphism is associated with a variety of autoimmune diseases, including type 1 diabetes, systemic lupus erythematosus, multiple scleral hyperplasia and rheumatoid arthritis [49]. There are also correlations with viral infections, such as COVID-19, WNV and EV71. IFIH1 rs9596310 is a risk factor for chronic hepatitis C virus (HCV) infection [50]. Both EV71 and HCV are positive single-stranded RNA viruses that may cause similar immune responses in the host. Our results showed that there was no significant difference in the frequency of the IFIH1 rs1990760 genotype among the mild, severe and healthy groups. At IFIH1 SNP rs1990760, the methylation level of IFIH1 in the TT genotype was significantly higher than that in the CC and CT genotypes. There was no correlation between the expression of IFIH1 and the genotype of IFIH1 rs1990760. The differences between our results and other studies may be related to the differences in case populations and grouping. Notably, we need to further expand the sample size to verify the correlation. A significant combined effect was observed between these two candidate SNPs, which showed that the risk of EV71 infection was increased with an increasing number of unfavorable DDX58 SNP rs3739674 GG and IFIH1 rs9596310 TT genotypes. Of course, we must discuss the potential limitations of our research. Little is known about the biological mechanism of the association between important SNPs and EV71 infection, and other factors need to be further considered. This aspect indicates that it is urgent to study the functional characteristics of these SNPs to verify our results.

In conclusion, this is the first report, to our knowledge, that elucidates DDX58 methylation patterns in EV71-HFMD. DDX58 expression and promoter methylation were associated with EV71 infection progression, especially in the severe EV71-HFMD group. Although future studies should expand the sample size and clarify the mechanism of how DDX58 influences EV71-HFMD, we herein provide a new perspective on the involvement of DDX58 in EV71-HFMD.

## Supporting information

**S1 STROBE Checklist. Checklist of items that should be included in reports of observational studies.**
(DOCX)

**S1 Table. Clinical data in 120 EV71-HFMD cases and 60 healthy control cases.**
(DOCX)

**S2 Table. DDX58 network functional analysis.**
(XLSX)

**S3 Table. IFIH1 network functional analysis.**
(XLSX)

**S1 Data. Clinical and Experimental Data in DDX58.**
(XLSX)

**S2 Data. Clinical and Experimental Data in IFIH1.**
(XLSX)

## Acknowledgments

The authors thank the doctors of the Department of Infectious Diseases of Xi'an Children's Hospital and Xi'an Jiaotong University Second Affiliated Hospital for their help with sample collection. The authors also thank the Shanghai Genesky Bio-Tech Center for Human Genetics Research for technical assistance with genotyping.

## Author Contributions

**Conceptualization:** Ya-Ping Li, Shuang-Suo Dang, Song Zhai.

**Formal analysis:** Ya-Ping Li, Chen-Rui Liu.

**Resources:** Hui-Ling Deng, Mu-Qi Wang, Yan Tian, Yuan Chen, Yu-Feng Zhang.

**Supervision:** Shuang-Suo Dang, Song Zhai.

**Writing – original draft:** Ya-Ping Li, Song Zhai.

**Writing – review & editing:** Ya-Ping Li, Chen-Rui Liu, Hui-Ling Deng, Mu-Qi Wang, Yan Tian, Yuan Chen, Yu-Feng Zhang, Shuang-Suo Dang, Song Zhai.

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
