## [Decision Letter · Decision Letter 0]

20 Jul 2021

Dear Dr. Zhai,

Thank you very much for submitting your manuscript "DNA methylation and single-nucleotide polymorphisms in DDX58 are associated with hand, foot and mouth disease caused by enterovirus 71" for consideration at PLOS Neglected Tropical Diseases. As with all papers reviewed by the journal, your manuscript was reviewed by members of the editorial board and by several independent reviewers. In light of the reviews (below this email), we would like to invite the resubmission of a significantly-revised version that takes into account the reviewers' comments. 

We cannot make any decision about publication until we have seen the revised manuscript and your response to the reviewers' comments. Your revised manuscript is also likely to be sent to reviewers for further evaluation.

Sincerely,

Johan Van Weyenbergh

Associate Editor

Jen-Ren Wang

Deputy Editor

Reviewer's Responses to Questions

**Key Review Criteria Required for Acceptance?**

**Methods**

-Are the objectives of the study clearly articulated with a clear testable hypothesis stated?

-Is the study design appropriate to address the stated objectives?

-Is the population clearly described and appropriate for the hypothesis being tested?

-Is the sample size sufficient to ensure adequate power to address the hypothesis being tested?

-Were correct statistical analysis used to support conclusions?

-Are there concerns about ethical or regulatory requirements being met?

Reviewer #1: in this study the authors are trying to find how DNA methylation at promoter, gene expression and SNPs at genes DDX58 and IFIH1 are regulated in to determine the the risk and severity of hand, foot, and mouth disease caused by enterovirus 71

(EV71-HFMD). They test 60 non patients and 60 mild and severe patients to test out the levels of DNA methylation at promoter and gene expression. The authors don't mention how they classify the patients as mild vs severe. More information in this regard would be helpful.

Reviewer #2: 1. Bonferroni corrections for multiple testing are needed, which should include any testing that was done on data that was not presented in the paper. In order to preserve the statistical significance of the main findings it may be necessary to clearly state a small number of hypotheses to be prospectively tested and present fewer minor findings. Maybe move them to supplementary data. 

2. How were the 60 control children recruited?

3. Were the 3 groups of children recruited simultaneously? What were the dates for the first and last recruited child in each group?

4. A table comparing the age/sex details of the children in each group would be helpful as well as first and last dates of recruitment.

5. Give more details of the patients who were excluded. E.g. How many? What were all the reasons?

**Results**

-Does the analysis presented match the analysis plan?

-Are the results clearly and completely presented?

-Are the figures (Tables, Images) of sufficient quality for clarity?

Reviewer #1: They show that DNA methylation on the promoter of DDX58 at some of the locations were higher in the severe dataset wrt to the mild and control. While some of the places showed a clear distinction between control and severe datasets with mild being in between the two, some of the regions like CpG 248, 168 and 179 had no difference between control and severe (figure 2). It would be more helpful if the authors provide individual data points in each of the 3 subsets to better understand the range of values ( in Figure 2,3,8). They also checked the promoter of IFIH1 but don't see a general effect though at very few points DNA methylation was higher in severe category as compared to control. here again authors claim that CpG86 was significantly different but the severe looks same in not lower than the control. (Both in Figure 3 and Figure 5). 

They compare the correlation between DNA methylation and gene expression for both DDX58 and IFIH1, and find an inverse correlation for many CpG methylation status with DDX58 expression. They show correlation in ALL people tested vs Patient population with not much change in the correlation values. It would make more sense to show correlation values in pateints and non patients(and not ALL) (in figure7). There was no significant correlation for IFIH1. 

The authors also checked the SNPs in DDX58 gene and find that DNA methylation and gene expression were correlated with the associated base in SNPs, but for IFIH1 gene the DNA methylation correlated well with the SNPs but not the gene expression. Again providing individual data points for this figure would be helpful in understanding the data better.

Reviewer #2: 1. The units of measurement should be stated on the y axis of figures and also in the figure legends.

2. How were the heat maps made? It would help to explain the meaning of the +1 and -1 values.

3. The word ‘compound’ in figure 7 is inaccurate.

4. I assume that the mean/median levels of GAPDH expression were the same in the groups? 

5. The SNPs were genotyped by two different techniques. Which technique was used for the results in the publication? Did the two techniques give the same results?

6. What were the results of the Hardy-Weinberg tests for each group?

7. What is the blue rectangle in the upper left of figure 1?

**Conclusions**

-Are the conclusions supported by the data presented?

-Are the limitations of analysis clearly described?

-Do the authors discuss how these data can be helpful to advance our understanding of the topic under study?

-Is public health relevance addressed?

Reviewer #1: In conclusion the authors claim that DDX58 expression and promoter methylation are associated with EV71 infection progression, which is collaborated by their data. Although their claim that the levels can be used to distinguish patients with severe disease may not hold true as the difference between control and severe patients in though significantly different in many cases was not hugely separated.

Reviewer #2: 1. In the conclusion of the Discussion the authors assume that the DDX58 parameters influence severe infection. It seems more likely that the infection influences the changes in methylation and expression. If DDX58 methylation was the cause of the severe infection (ie it preceded the infection) then the variance in the control group should be the same as the variance in the combined infected groups? In this data, high levels of DDX58 methylation are only seen after infection.

2. How biologically significant are these findings? It would be good to discuss this. How much of the variance (r squared) in DDX58 methylation is accounted for by clinical outcome? How much of the variance in DDX58 expression is accounted for by DDX58 methylation. How much of the variance in clinical outcome is accounted for by DDX58 expression?

3. How clinically significant are the findings? Are they likely to have any influence on patient management?

4. Etc is not a scientific expression

**Editorial and Data Presentation Modifications?**

Reviewer #1: Line 121: uppercase W in "written"

Reviewer #2: The standard of English is generally very good, except for the Author Summary which needs re-writing.

**Summary and General Comments**

Reviewer #1: (No Response)

Reviewer #2: The data is of interest and is a logical extension of previous work from the author's laboratory. There is too much data presented. Only present data that you are actually going to discuss.

-------------------
---

## [Decision Letter · Decision Letter 1]

8 Oct 2021

Dear Dr. Zhai,

Thank you very much for submitting your manuscript "DNA methylation and single-nucleotide polymorphisms in DDX58 are associated with hand, foot and mouth disease caused by enterovirus 71" for consideration at PLOS Neglected Tropical Diseases. As with all papers reviewed by the journal, your manuscript was reviewed by members of the editorial board and by several independent reviewers. In light of the reviews (below this email), we would like to invite the resubmission of a significantly-revised version that takes into account the reviewers' comments. 

We cannot make any decision about publication until we have seen the revised manuscript and your response to the reviewers' comments. Your revised manuscript is also likely to be sent to reviewers for further evaluation.

Sincerely,

Johan Van Weyenbergh

Associate Editor

Jen-Ren Wang

Deputy Editor

Reviewer's Responses to Questions

**Key Review Criteria Required for Acceptance?**

**Methods**

-Are the objectives of the study clearly articulated with a clear testable hypothesis stated?

-Is the study design appropriate to address the stated objectives?

-Is the population clearly described and appropriate for the hypothesis being tested?

-Is the sample size sufficient to ensure adequate power to address the hypothesis being tested?

-Were correct statistical analysis used to support conclusions?

-Are there concerns about ethical or regulatory requirements being met?

Reviewer #1: (No Response)

Reviewer #2: I have asked for more information about the study subjects - see comments 1, 2 and 3. Otherwise OK.

**Results**

-Does the analysis presented match the analysis plan?

-Are the results clearly and completely presented?

-Are the figures (Tables, Images) of sufficient quality for clarity?

Reviewer #1: (No Response)

Reviewer #2: The editor should decide whether the text of the figures is large enough to be legible in the published article. it looks a bit small to me.

**Conclusions**

-Are the conclusions supported by the data presented?

-Are the limitations of analysis clearly described?

-Do the authors discuss how these data can be helpful to advance our understanding of the topic under study?

-Is public health relevance addressed?

Reviewer #1: (No Response)

Reviewer #2: They address the public health relevance with ROC analyses, but probably overstate it. To be useful the analyses would need to be done prospectively at an early stage of infection before severe sequelae develop. It is possible that the changes in parameters they measure are a consequence of severe infection rather than the cause of it. They need to perform ROC analyses on lymphocyte and neutrophil data, which may have more predictive value and be far easier to measure. I see the value of this work, which I think has been carefully performed, as offering clues to disease pathophysiology that could be explored in further studies. For example I would be interested in knowing which subsets of blood leucocytes these changes occurred in and whether previous subclinical dengue virus infections or toxoplasmosis exposure influenced the chance of developing severe disease (unlikely but interesting if true).

They have a tendency to re-state the results in the discussion rather than discussing the results in the discussion. This makes it a bit laborious to read.

**Editorial and Data Presentation Modifications?**

Reviewer #1: (No Response)

Reviewer #2: Comments 4, 5 and 7 require data modifications. Comments 13 and 14 require small changes to statistics.

**Summary and General Comments**

Reviewer #1: The authors have addressed all the major comments and the revised manuscript is stronger and more clear.

Reviewer #2: This manuscript contains interesting data exploring the relationships between HFMD infection severity and DDX58 and IFIH1 promoter methylation, gene expression and SNP genotypes. There are still some improvements to be made.

1. Table S1 shows there were two windows for recruiting patients. Was the number of severe and mild cases the same in each window? 

2. The timing of the blood sample taking needs to be stated more clearly. Was the blood taken before or after the patients had ben classified into mild and severe groups? It would also be interesting to know the percentage of hospitalised patients that would normally be expected to develop severe disease.

3. Dengue virus infections have recently been reported in Xi’an and the peak of dengue infections overlaps HFMD. Is it possible that there was a different frequency of subclinical dengue infections between the mild and severe groups that might have influenced the study parameters?

4. Lines 226/227 say the 86 CpG has the highest methylation level in the severe group, but the data in the heat map (1C) and Table 2 show the opposite. This needs explanation.

5. The average methylation rectangles in heat map 1C clearly show that the mild group (brown) has higher methylation than the other groups (blue). However the data in figure 1D and in Table 2 do not show this. Why not?

6. Line 188. Say ‘the significance of differences between the groups was analysed ……’

7. CpG 179 had the highest methylation in the healthy group, not the severe group.

8. It is hard to imagine how a difference of between 7 and 9 methylated cytosines per 1000 cytosines (Table 1) can be of biological significance, although the data do show this difference is statistically significant. This needs to be discussed. 

9. Is the text in the figures large enough to be legible in the published article?

10. Does the low lymphocyte count in the severe group account for the methylation and mRNA transcription differences? That is, do lymphocytes have lower methylation levels and higher mRNA levels than non-lymphocytes and so when the lymphocyte count falls it appears as if methylation has increased and mRNA has decreased? 

11. Do methylation levels or mRNA levels make better predictions of disease severity than the lymphocyte count? If you were looking for a clinically useful prediction assay the lymphocyte count might be more convenient.

12. Are blood leucocytes, a complex mixture of many cell types, a good choice for these analyses? Surely it should be the cells infected by the virus. Maybe in future work you could determine which leucocytes these changes were occuring in. This would be relevant to understanding pathophysiology.

13. Since the distributions of the methylation and gene expression data were not normal, a Spearman rank order correlation coefficient is more appropriate.

14. Were the results of the Students t test adjusted for non-normal distributions?

15. Line 337. What is a virus cell?

16. Use the word ‘children’ instead of ‘kids’.
---

## [Decision Letter · Decision Letter 2]

14 Dec 2021

Dear Dr. Zhai,

We are pleased to inform you that your manuscript 'DNA methylation and single-nucleotide polymorphisms in DDX58 are associated with hand, foot and mouth disease caused by enterovirus 71' has been provisionally accepted for publication in PLOS Neglected Tropical Diseases.

Best regards,

Johan Van Weyenbergh

Associate Editor

Jen-Ren Wang

Deputy Editor

Minor changes needed: Line 95: parameters instead of reasons Line 219: remove 179 Line 267: Spearman's Line 273: Remove 246

Reviewer's Responses to Questions

**Key Review Criteria Required for Acceptance?**

**Methods**

-Are the objectives of the study clearly articulated with a clear testable hypothesis stated?

-Is the study design appropriate to address the stated objectives?

-Is the population clearly described and appropriate for the hypothesis being tested?

-Is the sample size sufficient to ensure adequate power to address the hypothesis being tested?

-Were correct statistical analysis used to support conclusions?

-Are there concerns about ethical or regulatory requirements being met?

Reviewer #2: No problems

**Results**

-Does the analysis presented match the analysis plan?

-Are the results clearly and completely presented?

-Are the figures (Tables, Images) of sufficient quality for clarity?

Reviewer #2: No Problems

**Conclusions**

-Are the conclusions supported by the data presented?

-Are the limitations of analysis clearly described?

-Do the authors discuss how these data can be helpful to advance our understanding of the topic under study?

-Is public health relevance addressed?

Reviewer #2: No problems

**Editorial and Data Presentation Modifications?**

Reviewer #2: Minor changes needed:

Line 95: parameters instead of reasons

Line 219: remove 179

Line 267: Spearmans

Line 273: Remove 246

**Summary and General Comments**

Reviewer #2: No comments

PLOS authors have the option to publish the peer review history of their article (what does this mean?). If published, this will include your full peer review and any attached files.

Reviewer #2: No

---

## [Editor Report · Acceptance letter]

4 Jan 2022

Dear Dr. Zhai,

We are delighted to inform you that your manuscript, "DNA methylation and single-nucleotide polymorphisms in DDX58 are associated with hand, foot and mouth disease caused by enterovirus 71," has been formally accepted for publication in PLOS Neglected Tropical Diseases.

Best regards,

Shaden Kamhawi

co-Editor-in-Chief

Paul Brindley

co-Editor-in-Chief
